# Hierarchical Zeolite Synthesis by Alkaline Treatment: Advantages and Applications

**Daniele S. Oliveira** [1], **Rafael B. Lima** [1], **Sibele B. C. Pergher** [1,*] and **Vinícius P. S. Caldeira** [2]

1    LABPEMOL, Institute of Chemistry, Federal University of Rio Grande do Norte, Natal 59078-970, RN, Brazil
2    LACAM, Chemistry Department, State University of Rio Grande do Norte, Mossoró 59610-210, RN, Brazil
*    Correspondence: sibele.pergher@ufrn.br; Tel.: +55-(84)-991936083

**Abstract:** Zeolites are of great interest to the scientific and industrial communities due to their interesting catalytic properties, such as high specific area, shape selectivity, and thermal and hydrothermal stability. For this reason, zeolites have been intensively studied and applied in several reactions of great industrial interest. However, the size of zeolite micropores may hinder the diffusion of bulky molecules in the pore system, limiting the use of zeolites in some reactions/applications that use bulky molecules. One way to address this limitation is to generate secondary porosity (in the range of supermicropores, mesopores and/or macropores) in such a way that it connects with the existing micropores, creating a hierarchical pore system. There are different hierarchical approaches; however, most are not economically viable and are complicated/time-consuming. Alkaline treatment has been highlighted in recent years due to its excellent results, simplicity, speed and low cost. In this review, we highlight the importance of alkaline treatment in the generation of secondary porosity and the parameters that influence alkaline treatment in different zeolitic structures. The properties and catalytic performance of hierarchical zeolites prepared by alkaline treatment are extensively discussed. It is expected that this approach will be useful for understanding how alkaline treatment acts on different hierarchical structures and will thus open doors to achieve other hierarchical zeolites by this method.

**Keywords:** hierarchical zeolites; alkaline treatment; porous materials; synthesis strategies; post-synthesis methods





## 1. Hierarchical Zeolites

Conventional zeolites contain only micropores with a maximum size of 2 nm in their structure. The presence of only micropores provides greater selectivity in these materials; however, the uniquely microporous structure prevents molecules (especially bulky molecules) from accessing the interior of the pores and finding the active sites of the zeolite. This limited diffusion can minimize the performance of zeolites, for example, by enabling the formation of coke that, in turn, promotes the deactivation of the zeolite.

Due to diffusion limitations and possible rapid deactivation when large molecules enter microporous zeolites, several recent studies have focused on obtaining hierarchical zeolites to improve the accessibility and catalytic efficiency of zeolitic materials.

To better understand the advantages and use of hierarchical zeolites, it is necessary to understand their structural composition in relation to porosity. Hierarchical zeolites exhibit intrinsic microporosity and secondary porosity, which can be in the range of supermicropores (0.7–2 nm), mesopores (2–50 nm), and macropores (>50 nm). This secondary porosity can be classified into porosity with a narrow or wide range of pore sizes. Therefore, hierarchical zeolites exhibit selectivity arising from their intrinsic microporosity, together with improved mass transport arising from the introduction of supermicropores, mesopores, or macropores into the zeolitic structure [1].

The introduction of secondary porosity into zeolites depends on the desired application of the materials. Generally, secondary porosity is in the range of mesopores. However, the generated pore size range can be controlled and/or influenced by the chosen hierarchical synthesis method. This secondary porosity can be understood by two distinct mechanisms. Additional supermicropores, mesopores or macropores can be generated in the zeolitic crystals, resulting in intracrystalline porosity, or they can also be generated by the agglomeration of nanometric zeolite crystals, resulting in intercrystalline porosity. In both cases, the obtained hierarchical zeolites exhibit improved mass transfer due to the greater accessibility caused by the additional porosity. Due to the benefits of enhanced accessibility, great interest has been devoted to obtaining hierarchical zeolitic materials [2,3].

According to some studies [3–6], hierarchical zeolites can be classified into three categories:

- Hierarchical crystals: formed by the combination of the intrinsic microporosity of the predominant crystal in the zeolite with additional intracrystalline porosity that may be in the range of supermicropores, mesopores, or macropores. In addition, the formation of intercrystalline macropores is possible through spaces between the crystals. This secondary porosity (supermicropores, mesopores or macropores) can be introduced by either direct synthesis or post-synthesis.
- Nanosized crystals: formed by decreasing the size of the zeolite crystals, which generally reach dimensions lower than 100 nm. In addition to the well-defined microporous system characteristic of zeolites, these materials have a system of intercrystalline supermicropores, mesopores or macropores caused by the agglomeration/packing of the crystals. This secondary porosity is obtained by direct synthesis.
- Supported zeolite crystals: the zeolite crystals are supported and/or dispersed in another material/support. The material obtained is not a pure zeolite but a material consisting of the zeolite micropore system and a system of mesopores and/or intercrystalline macropores in the support (the size range of the additional pores is determined by the support material). This secondary porosity is obtained by post-synthesis or assisted direct synthesis.

Hierarchical zeolites obtained by different synthesis methods exhibit accessibility advantages for various catalytic reactions when compared to conventional zeolites. These advantages contribute to increasing mass transfer, minimizing catalytic deactivation and increasing activity with respect to bulky substrates in several chemical reactions.

Therefore, new, effective processes for the formation of hierarchical zeolites have been widely studied. Essentially, there are two types of approaches to obtaining hierarchical zeolites: top-down and bottom-up approaches [7–10]. The first strategy consists of a post-synthesis procedure, in which an already established zeolitic structure is subjected to processes to generate secondary porosity. The second approach is a direct synthesis procedure, in which the secondary porosity derives from the formation of the zeolitic structure.

The main synthesis methods, based on hierarchical approaches, used to obtain hierarchical zeolites are presented in Table 1.

**Table 1.** Main methods for the synthesis of hierarchical zeolites.

| Approaches *Top-down* | Approaches *Bottom-up* |
| --- | --- |
|  | Hard templating [18] |
| Desilication [11,12] | Soft templating [19] |
| Dealumination [13] | Template-free [20] |
| Irradiation [14] | Dual templating with surfactant [21] |
| Recrystallization from mixed methods [15–17] | Zeolitization of materials [22] |
|  | Nanoparticle assembly [23] |

## 2. Synthesis of Hierarchical Zeolites by Post-synthesis Procedures

The top-down methods of dealumination, desilication, irradiation, and recrystallization are the main methods of secondary porosity formation by post-synthesis. These

methods have some advantages, such as applicability to different types of zeolites with different Si/Al ratios; low cost; high zeolitic crystallinity at the end of the process; and a high degree of secondary porosity. The main details of these post-synthesis methods are described below.

The dealumination method consists of the selective removal of aluminum atoms belonging to the zeolitic structure. This removal causes defects that are generated by the hydrolysis of Si-O-Al bonds [24–27]. Secondary porosity (usually mesopores) is introduced through the generation of vacancies caused by the removal of aluminum atoms [26–28]. However, this method exhibits some disadvantages, including the limitation of zeolites being rich in aluminum, difficulty in controlling the generated mesopores, poorly interconnected mesopores, and partial blockage of the pores (micropores and mesopores) caused by the deposition of amorphous material. In addition, the acidity of hierarchical zeolites can be significantly affected because the removal of aluminum atoms will reduce the number and strength of acid-active sites [9,25,29].

The irradiation method was developed by Valtchev [14]; in this method, macropores are introduced in a parallel orientation inside the zeolite crystals using uranium irradiation, and subsequently, etching with acid solution and washing with water are performed [1,30,31]. This process preserves the microporosity and crystallinity of the zeolite. Despite generating a uniform distribution of parallel macropores, this method has a disadvantage in the use of uranium to irradiate the zeolite crystals [1,30].

The desilication and recrystallization/restructuring methods are based on an alkaline treatment procedure. In desilication, silicon atoms are removed from the zeolitic structure, resulting in secondary porosity, usually in the range of mesopores [32–36]. Recrystallization/restructuring is characterized by the removal of atoms and subsequent reorganization of these atoms in the structure, which preserves the initial crystalline form and adds secondary porosity to make acid sites more accessible [37–40].

Desilication by alkaline treatment has become a topic of great relevance and interest in the preparation of hierarchical zeolites due to its success in obtaining zeolites with well-defined mesopores. This process introduces secondary porosity in a simple, fast, effective and low-cost way, which makes the method even more desirable. In addition, alkaline treatment can introduce secondary porosity while minimally affecting the acidic properties and microporous character of the zeolite, contributing to greater diversity in the application of these materials.

## 3. Alkaline Treatment

In recent years, alkaline treatment (a post-synthesis method usually involving NaOH solution) through desilication has become one of the most versatile procedures to generate mesoporosity in zeolites. Alkaline treatment can also lead to a reorganization in the zeolitic structure through the recrystallization/restructuring of the zeolite, generating secondary porosity and making acid sites more accessible.

The selective extraction of silicon from the structure by alkaline treatment, known as desilication or base leaching, is a top-down method widely used to prepare zeolites with hierarchical porosities. The controlled leaching of Si by $OH^-$ ions forms intracrystalline mesopores, which facilitates molecular access and diffusion in the active sites of the zeolite [41]. These changes bring enormous benefits in catalysis associated with greater activity, selectivity and/or lifespan. Figure 1 illustrates the desilication process.

Alkaline treatment can be performed by not only conventional electric heating but also microwave heating. Microwave radiation induces rapid and uniform heating and has selective interactions with certain reagents or solvents [41,42]. Both methods are effective in the production of zeolites with secondary porosity. However, the microwave method can be considered more efficient because it leads to the formation of mesopores within a short treatment time [43,44].

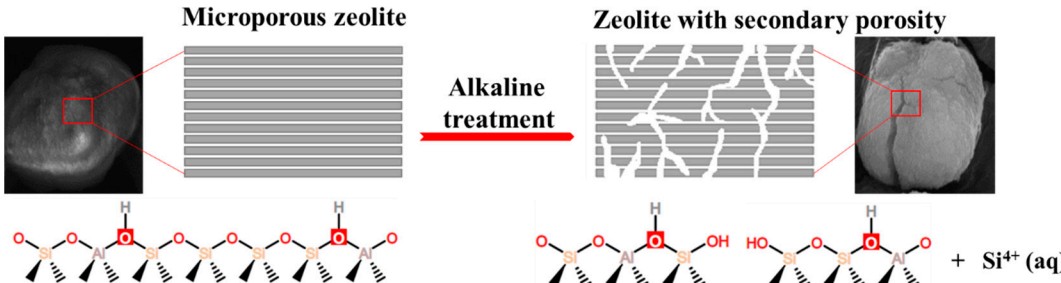

**Figure 1.** Method of desilication by alkaline treatment (SEM image of the ZSM-5 sample performed in TESCAN equipment, model MIRA3 FEG-SEM, magnification 2 μm).

The use of alkaline treatment (initially called caustic treatment) to improve the performance of zeolites in catalysis was patented by Dean Arthur Young and Yorba Linda in 1967 [45]. It was found that after treatment with solutions of alkali metal hydroxides, preferably NaOH, mordenite zeolite exhibited preserved crystallinity, a significantly increased benzene absorption capacity, and a threefold higher conversion to Pd/mordenite in oil and gas hydrocracking than standard mordenite zeolite (without alkaline treatment). Young and Linda speculated that the superior performance of the modified material could be due to better access to micropores.

In the patent "Caustic-treated zeolites" filed by Dean A. Young and Yorba Linda, 1968 [46], it was found that the properties of mordenite zeolite are improved by digestion with aqueous caustic solution (NaOH solution), in which a smaller proportion of structural silica is leached without significantly altering the crystal structure. It was found that caustic etching treatment increases the adsorption rate and effective capacity of zeolites, as well as their catalytic activity.

In 1977, Donald H. Rosback and Richard W. Neuzil [47] patented the use of an aqueous caustic solution (NaOH solution) in a precursor mass comprising zeolites X and Y in sodium form. They found that the treatment of this precursor mass under particular conditions produced an absorbent with increased capacity for olefins. The produced absorbent exhibits more efficient olefin separation due to its increased adsorption capacity and has a longer shelf life.

In 1979, Alan J. Rein, David D. Saperstein and Seemon H. Pines [48] patented a process for caustic washing of type 3A and/or 4A synthetic zeolites with an improved ability to eliminate acids. These materials can be used to prepare, for example, sodium 7-(2-thienylacetamido)-7-methoxy-3-carbamoyloxymethyl-3-cephem-4-carboxylate.

However, scientific articles on the modification of zeolite in alkaline media only began to be published approximately 25 years after the patent by Dean A. Young and Yorba Linda. In 1992, Dessau et al. [49] reported the dissolution of large ZSM-5 crystals in an attempt to identify Al gradients. They showed that treatment of ZSM-5 zeolite with an aqueous base ($Na_2CO_3$ solution) resulted in the partial dissolution of the sample with preferential removal of silicon. The treated zeolite had a lower silica/alumina ratio and exhibited a higher cation exchange capacity and higher catalytic activity.

In 1995, Le Van Mao et al. [50] analyzed the properties of zeolites Y, X, and ZSM-5 in alkaline media in more detail. It was concluded that treatment with aqueous sodium carbonate ($Na_2CO_3$) led to an increase in Al content and a higher ion exchange capacity without drastically altering the structure of the zeolites. This article reported the first nitrogen ($N_2$) adsorption/desorption isotherm of the mesoporous zeolite ZSM-5 obtained by alkaline treatment. Nevertheless, the main role of mesopores in increasing intracrystalline diffusion and/or access to micropore volume in reactions was not discussed.

In 1997, Čižmek et al. [51] focused on the dissolution mechanism of pentasyl zeolites with high Si contents (silicate-1 and ZSM-5 with different Al contents) in NaOH solution and confirmed the influence of aluminum on the dissolution kinetics.

However, it was only in 2000 that Ogura et al. [11] reported the importance of alkaline treatment and the remarkable porous changes it induced in ZSM-5 zeolite. They clearly demonstrated that the treatment of ZSM-5 in an alkaline solution (NaOH solution) drastically alters the morphology of ZSM-5, leading to the formation of mesopores with almost uniform size without destroying the microporous structure. They also observed by scanning electron microscopy (SEM) that after desilication, the morphology of ZSM-5 zeolite changed, with the appearance of cracks and holes, as shown in Figure 2.

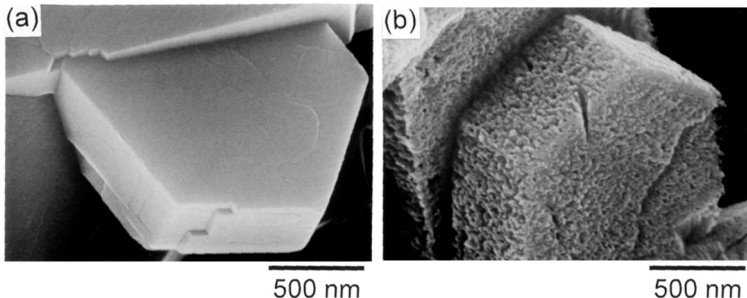

**Figure 2.** Desilication method: (**a**) SEM image of standard ZSM-5 and (**b**) SEM image of the ZSM-5 sample after alkaline treatment [11]. Copyright (2000), with permission from The Chemical Society of Japan.

Ogura et al. [11] performed $N_2$ adsorption/desorption at 77 K and observed a hysteresis loop in the desorption curve, a decrease in the microporous volume and an increase in the mesoporous volume, strongly suggesting the formation of mesopores in ZSM-5 zeolite. They concluded that the alkaline treatment of zeolite might be a promising procedure to create a uniform mesostructured material with zeolitic acidity.

Then, Groen, Pérez-Ramírez et al. [32,52–54] conducted several studies to investigate the potential of mesoporous zeolites (specifically ZSM-5) obtained by desilication. They established optimal conditions for the desilication method (0.2 mol $L^{-1}$ NaOH for 30 min at 65 °C) and reported that the crystallinity and acidity of the zeolite remained practically unchanged during alkaline treatment; they further stated that the versatility of this alkaline treatment opens new ways to improve the diffusion characteristics in zeolite-catalyzed applications.

Groen, Pérez-Ramírez et al. also stated that alkaline treatment selectively extracts silicon atoms from the zeolitic lattice. According to these authors, the obtained porosity seems to result from the preferential extraction of silicon from the structure due to hydrolysis in the presence of $OH^-$ ions. The aluminum in the structure controls the process of silicon extraction from the structure and makes the desilication process selective regarding the formation of intracrystalline mesopores. The presence of extra-lattice Al inhibits Si extraction and the formation of related mesopores; this is attributed to the re-alumination of extra-lattice Al species during alkaline treatment [32,52–56].

In recent years, the number of hierarchical zeolites prepared by alkaline treatment has increased significantly because this post-synthesis treatment method is effective, versatile and inexpensive and simply introduces secondary porosity into a wide variety of zeolitic structures. Table 2 shows some previously reported hierarchical zeolites prepared by alkaline treatment alone and by alkaline treatment combined with another approach.

It can be seen that different zeolites and treatments can be used to achieve hierarchical pores systems. Therefore, alkaline treatment can be used in different zeolites and can be easily extended to the industrial scale due to the simplicity and economy of the process as well as its compatibility with current technology. However, alkaline treatment can produce distinct effects on different zeolitic structures. This may occur due to differences in chemical and physical properties, such as composition, Si/Al molar ratio, symmetry, cell dimensions, structure, density, pore diameter and specific area. Therefore, they need to be better studied. In this article, we explore some of these properties in depth.

**Table 2.** Previously reported hierarchical zeolites prepared by alkaline treatment.

| Material Name | Type of Structure [a] | Pore Size (nm) | Morphology [b] | Secondary Porosity (nm) | Reference |
|---|---|---|---|---|---|
| ZSM-5 | MFI (3) | 0.51 × 0.55; 0.53 × 0.56 | Prismatic particles [c] | Intracrystalline, ~4 | 2000 [11] |
| ZSM-12 | MTW (1) | 0.56 × 0.60 | Clusters of small crystalline particles [c] | Intracrystalline, 15–20 | 2006 [57] |
| Mordenite | MOR (2) | 0.65 × 0.70; 0.26 × 0.57 | Ellipsoidal particles (4 × 2 × 2) | Intracrystalline [d] | 2007 [58] |
| Beta | *BEA (3) | 0.66 × 0.67; 0.56 × 0.56 | Truncated bi-pyramidal crystals [c] | Intracrystalline, ~3 | 2008 [59] |
| AlPO-16 | AST (0) | Apertures formed by 6MR | Poorly faceted crystals (0.1–0.2) | Intercrystalline, ~5 | 2008 [5] |
| Ferrierite | FER (2) | 0.42 × 0.54; 0.35 × 0.48 | Plate-like crystals (0.3–0.8 × 0.05–0.15) | Inter- and intracrystalline [d] | 2009 [60] |
| MCM-22 | MWW (2) | 0.40 × 0.55; 0.41 × 0.51 | Pellet-like crystals (1–1.5) | Inter- and intracrystalline [d] | 2009 [61] |
| ITQ-4 | IFR (1) | 0.62 × 0.72 | Rod-shaped crystals (beamlike) (1.5 × 0.2 × 0.2) | Intracrystalline, 4–10 | 2010 [62] |
| SSZ-35 | STF (1) | 0.54 × 0.57 | Agglomerates of small particles [c] | Not available | 2010 [63] |
| SSZ-13 | CHA (3) | 0.38 × 0.38 | Cubic crystals (10–15) | Intracrystalline, 2–10 | 2010 [64] |
| Zeolite Y | FAU (3) | 0.74 × 0.74 | Crystals [c] | Intracrystalline, ~2–10 and ~15–30 | 2010 [65] |
| Clinoptilolite | HEU (2) | 0.31 × 0.55; 0.41 × 0.41; 0.28 × 0.34 | Not available | Intracrystalline, ~11.7–17.8 | 2011 [66] |
| Silicalite-1 | MFI (3) | 0.51 × 0.55; 0.53 × 0.56 | Prismatic crystals (2.5) | Intracrystalline, ~10 | 2011 [67] |
| ZSM-22 | TON (1) | 0.46 × 0.57 | Nanorods (0.04 × 0.04 × 0.15) | Inter- and intracrystalline [d] | 2011 [68] |
| Nu-10 | TON (1) | 0.46 × 0.57 | Cylindrical particles (0.3–0.5 × ~0.05) | Intracrystalline, ~11.5 and ~20 | 2011 [69] |
| USY | FAU (3) | 0.74 × 0.74 | Octahedral crystals [c] | Intracrystalline, ~2.5–10 and ~2.5–20 | 2012 [70] |
| Zeolite X | FAU (3) | 0.74 × 0.74 | Octahedral crystals [c] | Intracrystalline, ~1000 | 2013 [71] |
| Zeolite L | LTL (3) | 0.71 × 0.71 | Clusters of crystalline particles [c] | Inter- and intracrystalline, ~3–50 | 2013 [72] |
| SSZ-74 | -SVR (3) | 0.55 × 0.57; 0.52 × 0.59; 0.52 × 0.56 | Rectangular crystalline particles [c] (5 × 8 × 0.55) | Intracrystalline, 5–20 | 2014 [73] |
| ZSM-23 | MTT (1) | 0.45 × 0.52 | Rod-like crystals (0.2–0.5) | Intracrystalline [d] | 2014 [74] |
| SSZ-33 | CON (3) | 0.64 × 0.70; 0.45 × 0.51 | Elliptical plates crystals (8 × 4 × 0.5) | Intracrystalline, 5–20 | 2015 [75] |
| EU-1 | EUO (1) | 0.41 × 0.54 | Clusters of small particles [c] | Intracrystalline, 10–20 and >50 | 2015 [76] |
| ZSM-11 | MEL (2) | 0.53 × 0.54 | Clusters of small crystalline particles (0.3–0.4) | Inter- and intracrystalline, 10–100 | 2015 [77] |

**Table 2.** *Cont.*

| Material Name | Type of Structure [a] | Pore Size (nm) | Morphology [b] | Secondary Porosity (nm) | Reference |
|---|---|---|---|---|---|
| IM-5 | IMF (3) | 0.55 × 0.56; 0.53 × 0.54; 0.53 × 0.59; 0.48 × 0.54; 0.51 × 0.53 | Square aggregates of flakes (irregular cubes) [c] | Inter- and intracrystalline, ~4–10 | 2018 [78] |

[a] Dimensionality in parenthesis. [b] Particle size in μm in parenthesis. [c] Particle size not available. [d] Range of secondary porosity not available.

### 3.1. Parameters That Influence Alkaline Treatment

Several parameters influence the generation of secondary porosity in hierarchical zeolites through alkaline treatment and need to be optimized, such as the Si/Al molar ratio, treatment time, temperature, alkaline agent concentration and zeolite structure. These parameters depend on the characteristics of the zeolitic structures that are used. In addition, the difficulty of controlling the size of the pores formed by this procedure represents an interesting field for investigation.

#### 3.1.1. Si/Al Molar Ratio

Alkaline treatment can generate new pores or expand existing micropores through the partial dissolution or reorganization/restructuring of the structure, which often depends on the Si/Al molar ratio of the starting zeolite.

Dessau, Valyocsik and Goeke [49] reported the selective removal of silicon in ZSM-5 zeolites with different Si/Al molar ratios (13, 14, 16, 22, 24, 29, 35, 38, 70, 74, 76, 81, 94, 124, 150 and 223). It was observed that ZSM-5 samples with Si/Al molar ratios between 70 and 223 partially resisted prolonged alkaline treatment (16 to 20 h under reflux). In all cases, partial dissolution with preferential Si removal was observed. The recovered zeolites were significantly enriched in aluminum, of which very little entered the solution phase.

In 1995, Le Van Mao et al. [50] studied the selective removal of Si (desilication) in ZSM-5, Y and X zeolites with Si/Al molar ratios of 19.5, 2.5 and 1.2, respectively. They concluded that as a general rule for efficient desilication, the higher the Si/Al ratio is, the greater the Si removal and the less basic the solution required for the treatment.

In 1995 and 1997, Čižmek et al. [51,79] analyzed the dissolution of zeolites of the pentasyl family with high silica content (silicatelite-1 and ZSM-5 with different Si/Al ratios) in NaOH solution. The experimental results showed that the dissolution of zeolites with high silica content is controlled by two essential reactions: a direct reaction caused by the breaking of Si-O-Si and/or Si-O-Al bonds due to the action of $OH^-$ ions in solution and a reverse process caused by the reaction between soluble species in the liquid phase or by reactions in/with the surface of the dissolved solid. The forward reaction rate decreased with increasing Al content in the ZSM-5 crystals, while the reverse reaction resulted in the formation of amorphous $SiO_2$ (ZSM-5) and/or different crystalline modifications of $SiO_2$ or even different hydrates of sodium silicate (silicatelite-1).

Groen, Pérez-Ramírez et al. [32,52,80–83] conducted extensive studies on the mesoporous ZSM-5 zeolite obtained by alkaline treatment. Commercial ZSM-5 zeolites with Si/Al ratios within the range of 15–1000 were used, and the usual desilication procedure was applied (0.2 mol $L^{-1}$ NaOH for 30 min at 65 °C). These studies showed that mesoporosity clearly depends on the Si/Al molar ratio of the ZSM-5 zeolites. Furthermore, they emphasized that a Si/Al molar ratio between 25 and 50 is ideal for obtaining appropriate mesopores under fixed treatment conditions, while the lowest Si/Al ratio (<25, high Al content) results in limited mesoporosity, and a higher Si/Al ratio (>50, high Si content) leads to extra macropores due to uncontrolled Si extraction. Thus, ZSM-5 zeolites with high Al contents are relatively inert to silicon extraction. This is because most Si atoms are stabilized near $AlO_4^-$ tetrahedra. Consequently, these materials exhibit a relatively low degree of silicon dissolution and limited mesoporosity, as shown in Figure 3 for ZSM-5 zeolite.

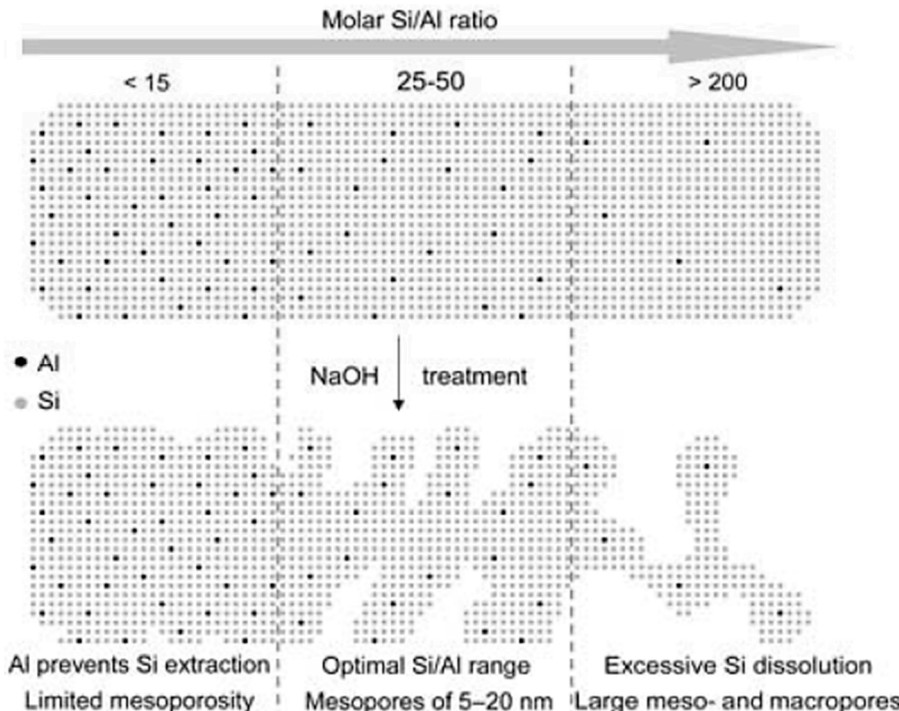

**Figure 3.** Influence of the Si/Al ratio on the desilication of ZSM-5 zeolite [52]. Copyright (2005), with permission from Wiley.

According to Groen et al. [52], the development of mesoporosity through Si extraction is fundamentally determined by the Si/Al ratio of the zeolite. Indeed, due to the negative charge of $AlO_4$ tetrahedra, the hydrolysis of the Si-O-Al bond in the presence of OH is impaired compared to the cleavage and relatively easy disruption of the Si-O-Si bond in the absence of neighboring aluminums. The number of aluminum atoms in the lattice is a determinant for the Si extraction process and, consequently, for mesopore formation [84].

According to Fernandez [85] and Verboekend and Pérez-Ramírez [86], virtually no aluminum is lost from the filtrate during alkaline treatment. From numerous analyses, the researchers found that in the process of creating mesopores, aluminum is removed from the structure and realuminated back on the outer surface of the zeolite. Therefore, the term "desilication" is not strictly accurate because although Si is leached in high amounts, both Si and Al are removed from the structure. This raises the question of whether realumination or the Al structure play the key role in the pore formation process. Other than aluminum being mainly a Lewis acid, very little is known about the nature and properties of aluminum deposited on the surface. Therefore, it is relevant to increase the understanding of these species, as they can significantly influence catalytic performance.

Verboekend and Pérez-Ramírez [67,86] recently extended the applicable Si/Al ratio range to 10–1000, which covers practically all compositions of ZSM-5 zeolite. For Al-rich zeolites (Si/Al = 10–20), alkaline treatment followed by subsequent acid washing removed amorphous Al-rich debris that blocked micro- and mesopores. For Si-rich zeolites (even silicic zeolites such as silicalite-1 and TS-1), deliberately added pore-directing agents (PDAs), such as $Al(OH)_4^-$ or tetrapropylammonium ($TPA^+$), are attracted to the surfaces of the zeolites, preventing excessive Si dissolution.

According to Verboekend [68] and Sadowska [87], intracrystalline mesoporosity is governed by the interaction of micro- and mesopores. However, it is also influenced by the Si/Al ratio of the structure. It was reported that alkaline leaching leads to the dissolution of Si and smaller amounts of Al species in the structure. However, most of these extracted Al species are able to realuminate at the surface of mesopores, resulting in a reduction in the Si/Al ratio of the hierarchical zeolite.

According to Verboekend, Vile and Pérez-Ramírez [88] and Tarach [89], the removal of silicon from the structure of some zeolites through alkaline treatment can cause the formation of large mesopores and a decrease in micropore volume and crystallinity. This shows that alkaline treatment, even under mild conditions, can affect the structural and acidic properties of some zeolites, such as beta zeolite (*BEA), and may be influenced by the content and position of Al atoms in the zeolite structure.

Zhang and Ostraat [2] performed pioneering work focused mainly on ZSM-5 zeolites and gradually applied post-synthesis alkaline treatment to other zeolite topologies, such as *BEA, FER, CHA, FAU, MTW, MWW, and MOR. The results revealed the challenges in obtaining the ideal mesoporosity with preserved microporosity/acidity for different zeolites through similar post-treatments with NaOH due to the differences in the stability of the Al structure and in the different crystallographies.

McGlone et al. [90] studied the alkaline treatment of ZSM-5 with $SiO_2/Al_2O_3$ ratios of 30 and 80. They concluded that desilication is more difficult at lower $SiO_2/Al_2O_3$ ratios due to repulsion by the negative charges on the ions imposed by the presence of Al in the structure and that it would be easier with $SiO_2/Al_2O_3$ ratios >50 due to uncontrolled desilication. In the case of $SiO_2/Al_2O_3$ ratio = 30, the rigid structure of the zeolite and the proximity of the Si and Al structures (Lowenstein's rule) prevent easy desilication to a certain extent.

These studies show that the secondary porosity generated by the alkaline treatment depends on the Si/Al molar ratio of the zeolite. A high Si/Al ratio can lead to the large formation of mesopores and extra macropores due to uncontrolled Si extraction. An average Si/Al ratio is considered ideal to obtain secondary porosity in the mesopore range. A low Si/Al ratio results in limited mesoporosity due to the greater amount of Al in the zeolite structure, which prevents greater dissolution. Al atoms have been shown to play a key role during alkaline treatment due to their influence on the stability of different crystallographic structures.

### 3.1.2. Alkaline Agents and Their Concentrations

The use of different alkaline agents and varying concentrations of solutions has been highlighted in recent years.

Suzuki and Okuhara [91] observed that alkaline treatment of ZSM-5 zeolite (Si/Al = 37) using lower-concentration NaOH solutions (0.05 mol $L^{-1}$ $dm^{-3}$, 90 $cm^3$) for 0.5–30 h formed supermicropores (approximately 1.8 nm) instead of mesopores. Thus, the concentration of the alkaline solution is an important factor controlling the zeolitic properties and pore size distribution, which can be directed to improve the catalytic functions of zeolites.

According to Groen [80], the removal of Si from the zeolite structure in an alkaline medium, for example, with NaOH and $Na_2CO_3$, is the simplest and most economical way to introduce mesopores into different types of zeolites without altering the acidic properties of the zeolitic structures. The preservation of the acidic properties of zeolites after desilication is related to the realumination of the aluminum species extracted, which promotes accessibility to available acid centers.

Wei and Smirniotis [57] investigated the influence of NaOH solutions with different concentrations (0.05, 0.1, 0.2 and 0.4 mol $L^{-1}$) and various treatment times and temperatures on the desilication of ZSM-12 zeolite with different Si/Al ratios. The concentration of the NaOH solution was considered the most dominant factor affecting the desilication method.

Pioneering studies used the alkaline agents NaOH and $Na_2CO_3$ for desilication. However, studies have been conducted with other alkaline agents, such as the study performed by Groen, Moulijn and Pérez-Ramírez [83] using inorganic bases such as KOH and LiOH. These bases were shown to be less effective than NaOH in the development of mesopores.

According to Serrano, Escola and Pizarro [30], the most common and most studied desilication procedure involves the treatment of zeolite with 0.2 mol $L^{-1}$ NaOH solution for 30 min at 65 °C using a zeolite-to-solution ratio of 33 g $L^{-1}$. Under these conditions,

silica is preferentially removed from the crystal structure, which gives rise to mesoporosity, causing a decrease in the Si/Al atomic ratio of the desilicated zeolite.

Gackowski et al. [36] studied the mild alkaline treatment of ultrastable zeolite Y with a high silica content (Si/Al = 31) using 0.05 and 0.2 mol $L^{-1}$ ammonia solutions and observed a major impact on the structure and properties of the desilicated samples, even in highly diluted ammonia solution (0.05 mol $L^{-1}$). The amount of silica extracted from the zeolite crystals under these conditions is quite low, leading, however, to significant structural changes in the solids. Thus, a high degree of amorphization was observed, as well as simultaneous changes in Al status and the creation of a high volume of mesopores. It was shown that the samples treated with dilute ammonia solutions exhibited short-range order associated with high Brønsted acidity. According to the authors, from an economic point of view, the treatment of zeolites with inexpensive ammonia solutions is more convenient than a hierarchical zeolite synthesis route based on expensive alkaline solutions, such as TPAOH.

According to Dai [92], in the alkaline treatment of the zeolite H-ZSM-5 using NaOH solution with different concentrations (0.1, 0.3, 0.5, 0.7 and 0.9 mol $L^{-1}$), when the concentration of the alkaline solution was low (<0.5 mol $L^{-1}$), the microporous area and microporous volume decreased slightly with the formation of mesopores. This indicates that moderate alkaline treatment increases the mesoporosity of the zeolite and preserves the micropore structure. However, the microporosity was severely destroyed after treatment with high concentrations (>0.5 mol $L^{-1}$). In addition, the mesoporous volume decreased after desilication with 0.9 mol $L^{-1}$ alkaline solution. This may be a result of the collapse of the zeolite channels caused by the dissolution of zeolitic crystals.

According to Zhang [93], the hierarchical zeolite H-ZSM-5 with Si/Al ratio = 20 was synthesized by alkaline treatment using aqueous solutions of LiOH, NaOH, KOH and CsOH at the same concentration (0.2 mol $L^{-1}$). As expected, there was an increase in surface area and mesoporous volume after LiOH, NaOH and KOH treatment, which indicates that some of the micropores were destroyed and intracrystalline mesopores were formed. This indicates that $OH^-$ ions can easily attack the internal siloxane groups within the zeolite channels. For CsOH-treated H-ZSM-5, the surface area did not increase dramatically due to more severe alkaline treatment (lower relative crystallinity), and its micropore surface area did not decrease, possibly due to crystal fragmentation (intracrystalline mesopores are covered by crystal fragments). This result indicates that treatment with CsOH is an effective and easy way to induce mesoporosity while maintaining microporosity. The hierarchical catalyst H-ZSM-5 treated with CsOH exhibited not only adequate acidity but also open interconnected mesopores and a smaller crystal size, resulting in greater catalytic activity and stability due to the presence of shorter diffusion paths (prolonging the usefulness of the catalyst). To better understand the main mechanism of enhanced diffusion, it is schematically demonstrated in Figure 4.

According to Tang [94], a series of hierarchical H-ZSM-5 zeolites were prepared by alkaline treatment using varying concentrations of NaOH solutions (0.1–0.5 mol $L^{-1}$) and the following solutions: 0.3 mol $L^{-1}$ $NaAlO_2$, 0.3 mol $L^{-1}$ $Na_2CO_3$ and 0.3 mol $L^{-1}$ TPAOH. The acidic properties of H-ZSM-5 zeolite were less affected after treatment with NaAlO2, $Na_2CO_3$ and TPAOH than with NaOH. Fewer mesopores were introduced into the H-ZSM-5 treated with $Na_2CO_3$ than that treated with NaOH. Treatment with TPAOH did not have a significant effect on the introduction of mesoporosity into H-ZSM-5 because TPAOH acted as a template, helping to repair the crystal structure of the zeolite. Among the alkaline treatment conditions employed, 0.3 mol $L^{-1}$ NaOH resulted in the best production of aromatic hydrocarbons.

Tanaka et al. [95] performed an alkaline treatment of zeolite H-ZSM-5 with $Na_3PO_4$ and NaOH, followed by acid treatment with $H_3PO_4$. H-ZSM-5 was kinetically more stable when treated with $Na_3PO_4$ than with NaOH at the same alkalinity. Thus, as in the treatment with NaOH, the yield and crystallinity decreased gradually with time under $Na_3PO_4$ treatment, and the volume of mesopores increased. In contrast to NaOH treatment,

phosphorus species were introduced into the products by $H_3PO_4$ treatment. Acid treatment using $H_3PO_4$ was combined with NaOH treatment. In contrast to the alkaline treatments, the crystallinity, micropore volume and surface area increased slightly with $NaOH/H_3PO_4$ treatment time. While the yield of the solid product decreased, the Si/Al ratio increased, indicating the dealumination of the structure with $NaOH/H_3PO_4$. As in the treatment with $H_3PO_4$, phosphorus species were introduced into the products by treatment with $NaOH/H_3PO_4$.

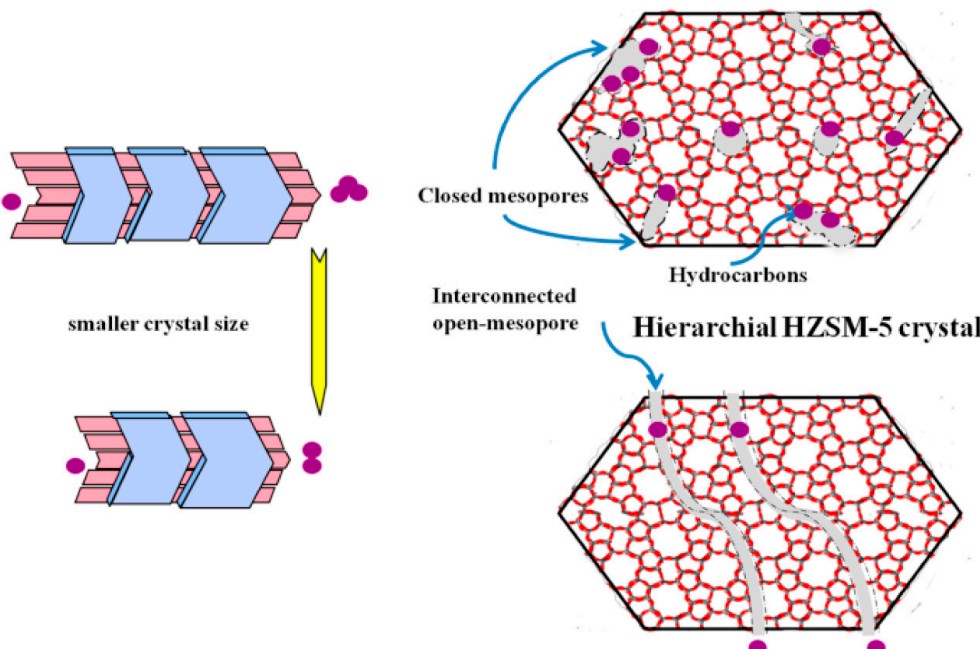

**Figure 4.** Schematic illustration of the main mechanism of enhanced diffusion [93]. Copyright (2018), with permission from MDPI.

The choice of the alkaline agent and the concentration of the alkaline solution are important factors controlling the zeolitic properties and pore size distribution. According to published studies, the more basic the alkaline agent is, the greater the volume and size of pores formed, and conversely, the lower the concentration is, the lower the volume and size of pores formed. By varying the concentration and type of the alkaline agent, the size and volume of the secondary porosity, the crystallinity and the acidity of the obtained zeolite can be controlled to some extent.

### 3.1.3. Templates as a Secondary Porosity Driver

The possibility of controlling the pore size in alkaline treatment with different pore-directing agents (PDAs) is currently being intensively researched.

Studies seeking to better control desilication introduced an additional compound into the solution to finalize the process, especially for less stable types of zeolites that are difficult to handle in NaOH solution. It was observed that the PDAs commonly used in the synthesis of zeolites could also be used as alternative compounds. Inspiration began with the use of aqueous solutions of tetraalkylammonium hydroxides (TAA, TPAOH, TBAOH, TMAOH) with the base medium [41].

Figure 5 compares alkaline treatment in conventional leaching with PDA treatment. The zeolite treated with PDAs exhibited the formation of more controlled mesopores.

Perez-Ramirez et al. [96] studied the desilication method involving NaOH treatment using ZSM-5 zeolite in the presence of quaternary ammonium cations. They found that these PDAs, such as $TPA^+$ and tetrabutylammonium hydroxide ($TBA^+$), act as a moderator of pore growth in zeolites by the extraction of silicon aided by $OH^-$, largely protecting the structure of zeolites during desilication and improving transport and catalytic performance.

This protective effect is not seen when cations capable of penetrating zeolite micropores, such as tetramethylammonium ($TMA^+$), are used.

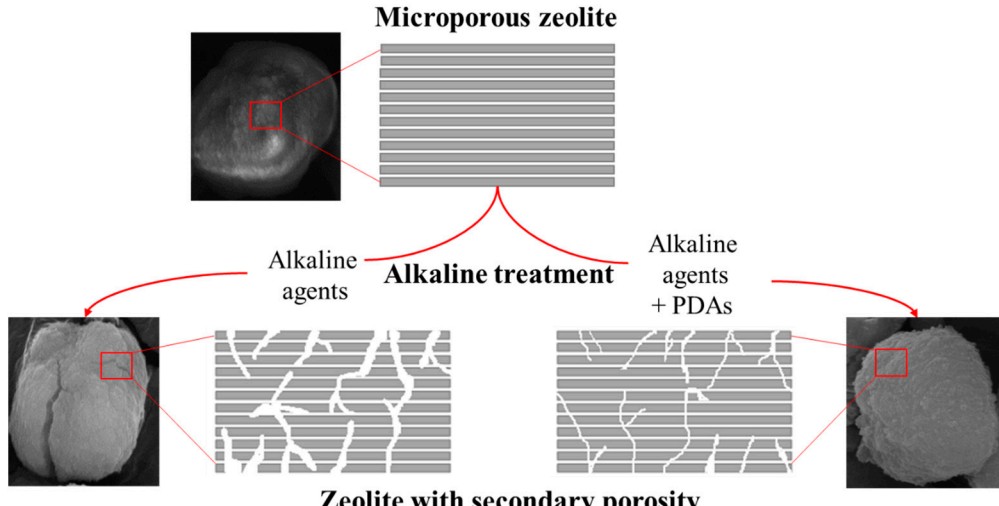

**Figure 5.** Illustration of alkaline treatment with and without PDAs.

Holm, Hansen and Christensen [97] investigated a "one-pot" desilication and ion exchange procedure, in which a subsequent ion exchange step that protonates the zeolite is avoided. This is possible because the $TMA^+$ charge compensation ions in the zeolite structure decompose during calcination to give protons. Additionally, according to the authors, the desilication of beta zeolite in a protonated form, i.e., without sodium in TMAOH, results in the limited formation of mesopores. This result indicates that $Na^+$ is required for efficient hydrolysis of the Si-O-Si bond, similar to the mineralization capacity during synthesis. Another potentially important consideration is the fact that in the absence of sodium during desilication, the structural charge will be compensated by only the $TMA^+$ ion, which can increase diffusion restrictions and thus prevent or simply delay the formation of pores.

Verboekend and Pérez-Ramirez [67] investigated the role of PDAs in the introduction of hierarchical porosity into silicalite-1 in an alkaline medium. They observed that the pore-directing role is not exerted directly by the trivalent cation metals of the structure but by species on the external surface of the zeolite. The inclusion of metal complexes ($Al(OH)_4^-$, $Ga(OH)_4^-$) and tetraalkylammonium cations ($TMA^+$, $TPA^+$) in the alkaline solution led to distinct mesopore surface areas and pore sizes centered in the range of 5 to 20 nm. All the aluminum partially integrated into the zeolite gave rise to Lewis and Brønsted acidity.

Verboekend and Pérez-Ramirez [67] also proposed a desilication model relating the affinity of the zeolite surface with the PDA and its mesopore formation efficiency (Figure 6). The illustration shows that the optimal formation of intracrystalline mesopores by controlled silicon leaching depends on a balance between the affinity of the PDA with the zeolite surface and the desilication of the zeolite crystal (the amount of PDA is exactly the same in the three scenarios). Evidently, when PDA does not show an affinity for (or is repelled by) the zeolite surface, there is no protection, which results in excessive dissolution (standard alkaline treatment). At the opposite extreme, when the affinity for the zeolite is very strong, as in the case of $TMA^+$, the surface is overprotected, and the dissolution process is completely inhibited, leading to a high yield of the solid and a lower formation of mesopores. In the medium, which is representative of $Al(OH)_4$ and $TPA^+$, the optimal balance between affinity for the zeolite and dissolution results in the formation of mesopores. In this case, $TPA^+$ proved to be the most effective PDA for the generation of mesopores in all silicic zeolites (causing partial protection of the zeolite surface).

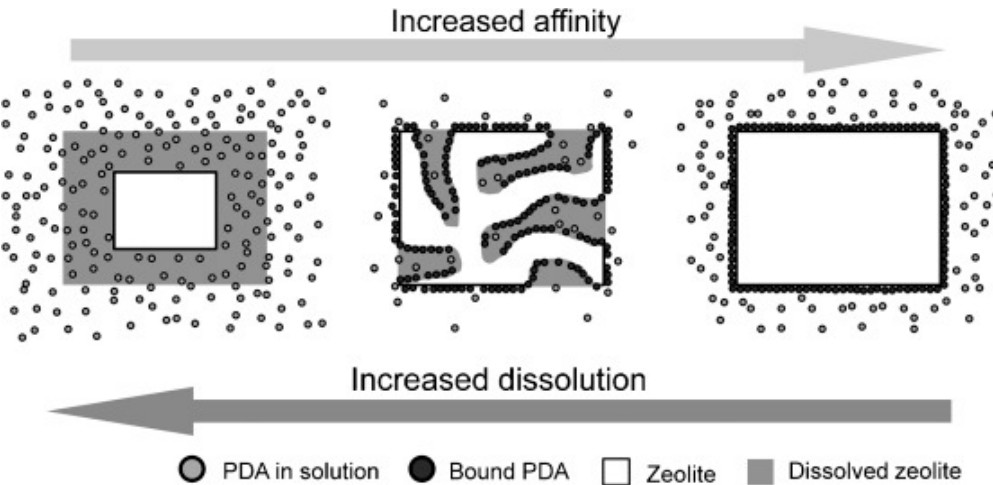

**Figure 6.** Illustration of how the affinity of PDA to the zeolite surface influences the rate of zeolite dissolution and the formation of mesopores [67]. Copyright (2011), with permission from Wiley.

Sadowska, Gora-Marek and Datka [98] investigated the acidity of ZSM-5 zeolite by desilication using NaOH and a mixture of NaOH/tetrabutylammonium hydroxide (TBAOH) with various concentrations, as well as different ratios of NaOH to TBAOH. It was demonstrated that the concentration of Brønsted acid centers increased as a result of the decrease in Si/Al due to desilication in dilute basic solutions (0.1 and 0.2 mol L$^{-1}$). Desilication in more concentrated bases (0.5 mol L$^{-1}$) resulted in the partial destruction of the zeolite, leading to the generation of weakly acidic protonic centers (groups other than SiOHAl). Studies have shown that desilication, which produces mesopores, improves the accessibility of acid centers. This effect was more apparent when desilication was performed with a NaOH/TBAOH mixture.

Sadowska et al. [87] found that NaOH/TBAOH mixtures produced mesopores with greater surface area and pore volume and smaller diameters in ZSM-5 zeolite than did NaOH. It is assumed that in the presence of TBAOH, a greater number of narrower pores are formed, or the narrow pores penetrate the zeolite crystal more deeply.

Tarach et al. [89] investigated the alkaline treatment of beta zeolite (Si/Al = 22) with NaOH and NaOH/TBAOH and found that desilication with NaOH/TBAOH resulted in more uniform intracrystalline mesoporosity with the formation of narrower mesopores while preserving the degree of crystallinity, resulting in catalysts with the most appropriate acidity and with better catalytic performance.

According to Raad [99], an alkaline treatment of H-ZSM-5 zeolite using NaOH with or without TBAOH leads to the formation of more or less structured intracrystalline mesopores. The presence of TBAOH allows the shaping of mesopores. Alkaline treatment preserves most of the Brønsted acid sites without any change in their concentration. A small number of new types of Lewis acid sites appear, particularly in the presence of TBAOH.

According to Verboekend, Vilé and Pérez-Ramírez [88], the desilication efficiency of a tetraalkylammonium cation strongly depends on its affinity with the zeolite surface. Nevertheless, only common quaternary ammonium cations used in the synthesis of zeolites have been explored for this purpose, for example, TPA$^{+}$ and TMA$^{+}$. Most likely, other molecules used in the preparation of zeolites exert the same or even a superior effect. For example, cetyltrimethylammonium (CTA$^{+}$), often used as a secondary template (mesopore template) during the synthesis of hierarchical zeolites, should exhibit a distinct effect on zeolites under alkaline conditions.

Additionally, according to Verboekend, Vilé and Pérez-Ramírez [88], the use of PDAs in NaOH leaching is a generic procedure to introduce intracrystalline mesoporosity into USY and beta zeolites while preserving the intrinsic properties of the zeolite, for example, microporosity, crystallinity and composition. It was found that a wide variety of quaternary ammonium cations and amines positively influence the desilication of these structures and

that their impact depends on their charge and size. The type and concentration of PDAs affect the external (mesopore) surface and the size of mesopores, in addition to minimizing amorphization. They showed that TPA$^+$ produces mesoporous solids with greater zeolitic properties, while CTA$^+$ gives rise to the reassembly of dissolved species.

According to Zhang and Ostraat [2], a series of cationic, nonionic and anionic surfactants (PDAs) were combined with NaOH to optimize the formation of mesopores. An effective PDA for desilication requires a cationic charge and a mixture of alkyl compounds in the range of 10–20 carbon atoms, such as TPA$^+$ (tetrapropylammonium) and CTA$^+$ (cetyltrimethylammonium) cations. The use of TPA$^+$ results in the formation of highly mesoporous zeolites that retain intrinsic zeolitic properties; however, the use of CTA$^+$ facilitates the reassembly of dissolved species during alkaline treatment. Thus, both cations are ideal modifiers for desilication in terms of creating mesoporosity and preserving microporosity.

According to Ying and Garcia-Martínez [37] and Chal [38], alkaline treatment can also lead to the restructuring of the zeolite, generating secondary porosity and making acid sites more accessible.

Wang et al. [100] reported a new simple synthesis method to generate extra porosity by the recrystallization of MOR zeolite from NaOH solution in the presence of a mesoporous director. MOR/MCM-41 was successfully synthesized with more accessible acid sites.

Chal et al. [38] studied the recrystallization of zeolite Y using an organic base (TMAOH) in combination with a cationic surfactant (CTAB). The formation of mesoporosity was observed in the zeolitic structure, and the initial crystalline form was preserved.

Yoo et al. [101] presented a method for the preparation of mesoporous zeolite by a combination of desilication and reassembly methods, adopted from the concept of pseudomorphic transformations. Dissolved species containing silicates, aluminosilicates and fragments of ZSM-5 crystals can be deposited back into the zeolite structure by surfactant molecules (such as CTAB) through the formation of micelles under hydrothermal conditions (Figure 7). The final products prepared under specific alkaline conditions exhibited a bimodal mesopore size distribution (3 and 10–30 nm), increased external surface area and well-preserved crystallinity. The reassembly of the dissolved species by surfactant micellization produced small mesopores (3 nm), while desilication generated larger mesopores (10–30 nm).

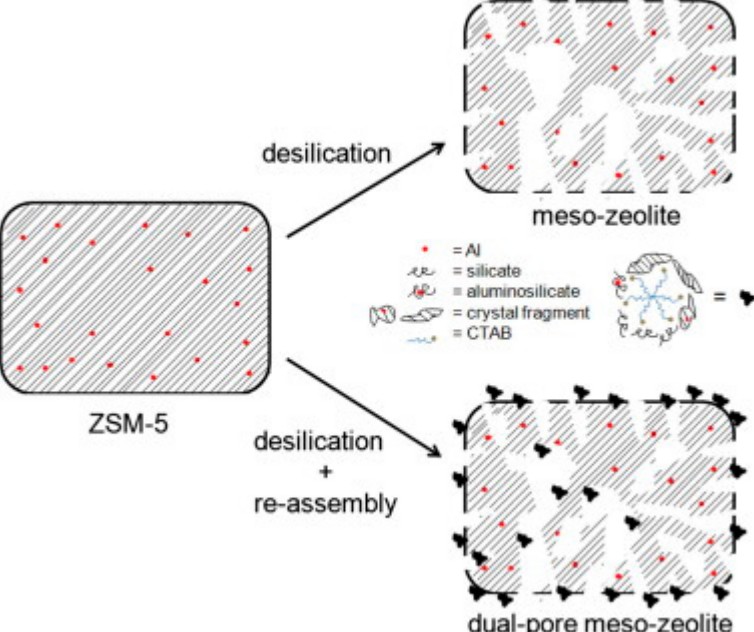

**Figure 7.** Schematic illustration of the preparation of mesoporous zeolite by means of desilication (upper right) and desilication/reassembly (lower right) [101]. Copyright (2012), with permission from Elsevier.

Garcia-Martínez et al. [15,37] described a new surfactant-based technique that allows precisely controlled mesoporosity to be introduced into a wide range of zeolite crystals, for example, FAU, MOR and MFI with various Si/Al ratios, while maintaining the chemical and physical properties of the zeolites. They suggested that the introduction of controlled mesoporosity in zeolites may occur through a surfactant-assisted crystal rearrangement mechanism. In summary, zeolite Y with well-controlled intracrystalline mesoporosity was prepared by means of surfactant-assisted synthesis (CTAB). According to the authors, this was the first time that well-controlled mesoporosity was introduced in Y zeolites with low Si/Al ratios, which are relevant for catalytic cracking.

Ivanova et al. [102] suggested a mechanism for the recrystallization of mordenite zeolite (Figure 8). According to the authors, in the first step of the process, Si-O-Si bonds are broken in an alkaline medium by desilication. The formation of negatively charged sites during desilication and ion exchange (sodium cations penetrate the interior of the crystals, and all zeolite protons and SiOH groups undergo rapid ion exchange with $Na^+$) stimulates the diffusion of $CTA^+$ cations attracted by negative charges in the intercrystalline and intracrystalline spaces of the zeolite. Hydrothermal treatment at 150 °C promotes the condensation of the species formed during desilication around the micelles, which results in the formation of fragments with uniform mesopores. The formation of such fragments occurs both inside the zeolite crystals and on the external surface of the crystallites. The resulting material is a zeolite with bimodal porosity (micro- and mesopores).

Silva, Ferracine and Cardoso [103] evaluated the effects of different concentrations of NaOH and the surfactant cetyltrimethylammonium bromide ($CTA^+$) on the textural, chemical and morphological characteristics of USY zeolite (Si/Al ratio = 15). The generation of mesoporosity in USY zeolite was enabled by the simultaneous presence of the surfactant and alkaline solution. Among the parameters studied, the concentration of the alkaline medium had the greatest influence on the textural properties of the zeolite. The presence of $CTA^+$ cations was fundamental in the process since the cations hindered the attack of hydroxyl groups ($OH^-$), preventing the dissolution of zeolite crystals during the treatment.

Gorzin et al. [104] evaluated the generation of mesoporosity in highly silicon-rich H-ZSM-5 zeolites (Si/Al ratio = 400) by a two-step approach: alkaline treatment using $NaAlO_2$ and $NaAlO_2$/TPAOH in different proportions, followed by washing treatment with acid. The results showed that the porosity of the desilicated samples was blocked mainly by deposits of sodium aluminate and silicon-containing debris. After a subsequent washing step with hydrochloric acid, the blocking species were removed, which resulted in improved mesoporosity. It was found that alkaline treatment in $NaAlO_2$/TPAOH solution followed by acid washing leads to the formation of narrow and uniform mesoporosity without severely destroying the crystal structure. The results showed that the desilication process leads to the considerable development of mesoporosity, while acid-washing treatment mainly influences the acidity of the catalyst. Therefore, the combination of alkali-acid treatment leads to the formation of a hierarchical H-ZSM-5 catalyst with a customized pore architecture and acidic surface properties.

Al-Ani et al. [105] concluded that surfactant species play a key role in the formation of regular intracrystalline mesopores and in the protection of the zeolite structure against desilication and excessive dealumination during the mesostructuring process; this process is in contrast to the treatment of zeolites by recrystallization, which can lead to the degradation and amorphization of the zeolite structure, generating a high degree of mesoporosity. According to the authors, zeolite treatment in the presence of an organic surfactant in a basic medium results in a much more controllable mesostructuring procedure than is possible with severe leaching using inorganic basic solutions (e.g., NaOH solution) without a surfactant, which generally leads to the significant degradation of the zeolite structure.

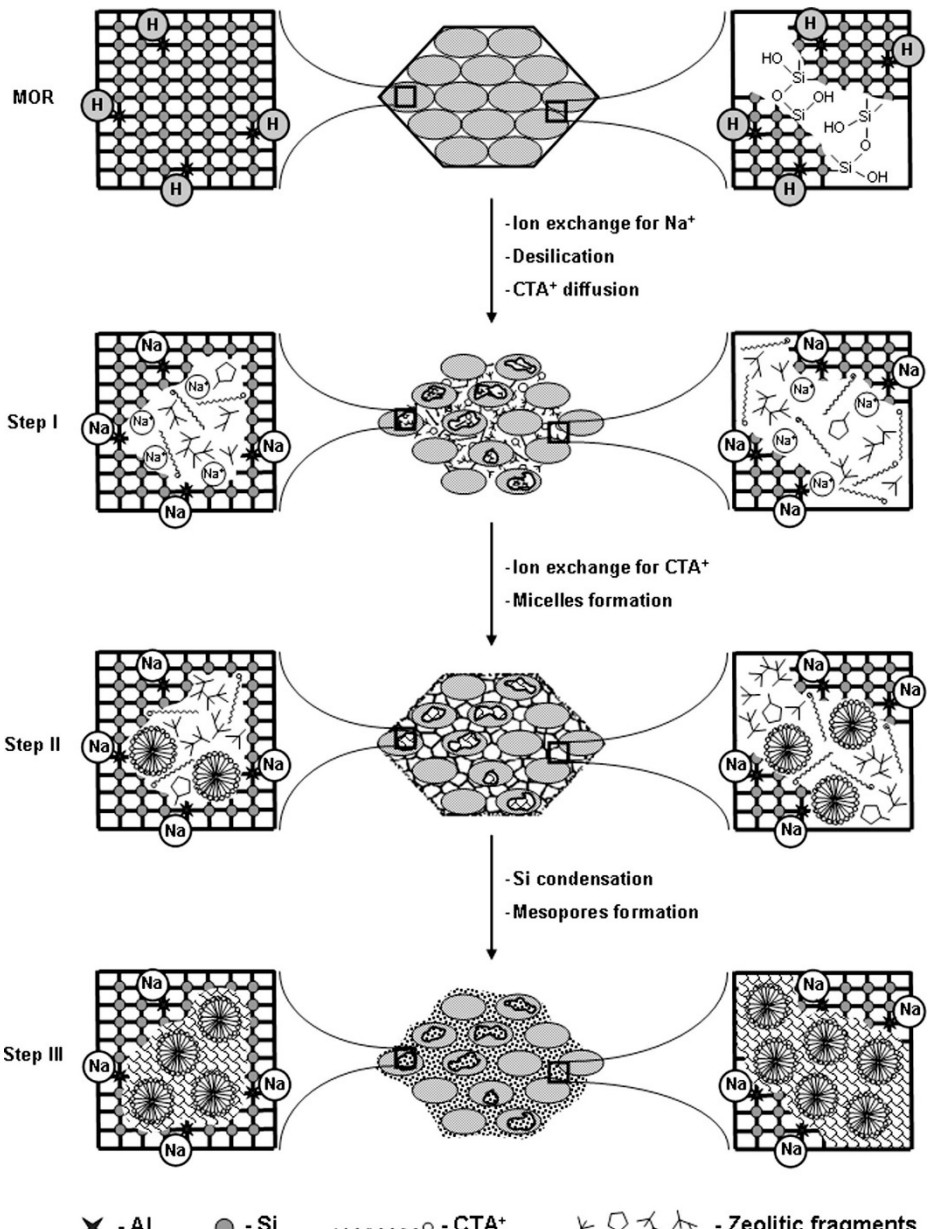

**Figure 8.** Proposed mechanism of the recrystallization of mordenite zeolite [102]. Copyright (2014), with permission from Elsevier.

### 3.1.4. Morphology and Crystal Size

According to Groen [52], information on how the porosity is distributed among the crystals and on the effect of crystal size is essential to rationally plan the potential applications of zeolites treated with an alkaline solution. It was shown that the controlled desilication of large and small crystals of ZSM-5 could modify and utilize the impact of Al zoning on the development of mesoporosity. The authors concluded that the Al gradient affects Si extraction throughout the zeolite particle and that the introduction of mesopores by desilication into large crystals may be inefficient due to Al zoning. Thus, for samples with a high degree of aluminum zoning, hollow crystals with an empty core and intact shell can be created.

Dessau, Valyocsik and Goeke [49] observed the nonuniform dissolution of large crystals of ZSM-5 (5–10 μm) after prolonged treatment with high concentrations of alkaline solution. This was speculatively attributed to the nonuniform concentration of Al in the crystals.

Verboekend et al. [68] suggested that the limited mesopore specific area and low desilication efficiency obtained in the alkaline treatment of ZSM-22 zeolite (one-dimensional) and ferrierite zeolite (two-dimensional) must be related to the morphology of the crystals. The introduction of mesoporosity into ZSM-22 crystals is not directly due to the unique characteristics of this zeolite: its nanorod (needle-like) morphology, unidimensional ellipsoidal microporous system and uneven distribution of Al. It was demonstrated that the desilication efficiency of ZSM-22 nanorods and ferrierite platelets is relatively low compared to that of ZSM-5, most likely due to the crystal morphology of the former two zeolites that facilitates the creation of intercrystalline mesoporosity.

Van Laak et al. [106] performed alkaline treatment on ZSM-5, ZSM-12 and beta zeolites. The ZSM-5 samples consisted of small crystallites between 20 and 100 nm in size that were agglomerated into larger particles of approximately 1 μm. SEM analysis (Figure 9A–C) indicated that the mesopores were intercrystalline and intracrystalline. Additional experiments on ZSM-12 yielded similar results and showed that the morphology of the standard zeolite (Figure 9D–F), i.e., crystallite/particle size, determines the amount of added mesoporosity, in which smaller crystallites give rise to larger pores. They also performed alkaline treatment on beta zeolite with 5 μm particles, where the deagglomeration of the particles was observed (Figure 9G–I). All of the tested zeolites (ZSM-5, ZSM-12 and beta) consisted of small crystallites (30–200 nm) that were agglomerated into larger particles between 1 and 5 μm. Intercrystalline mesopores were formed for all zeolites, but the treatment was more effective for zeolites with small crystallites.

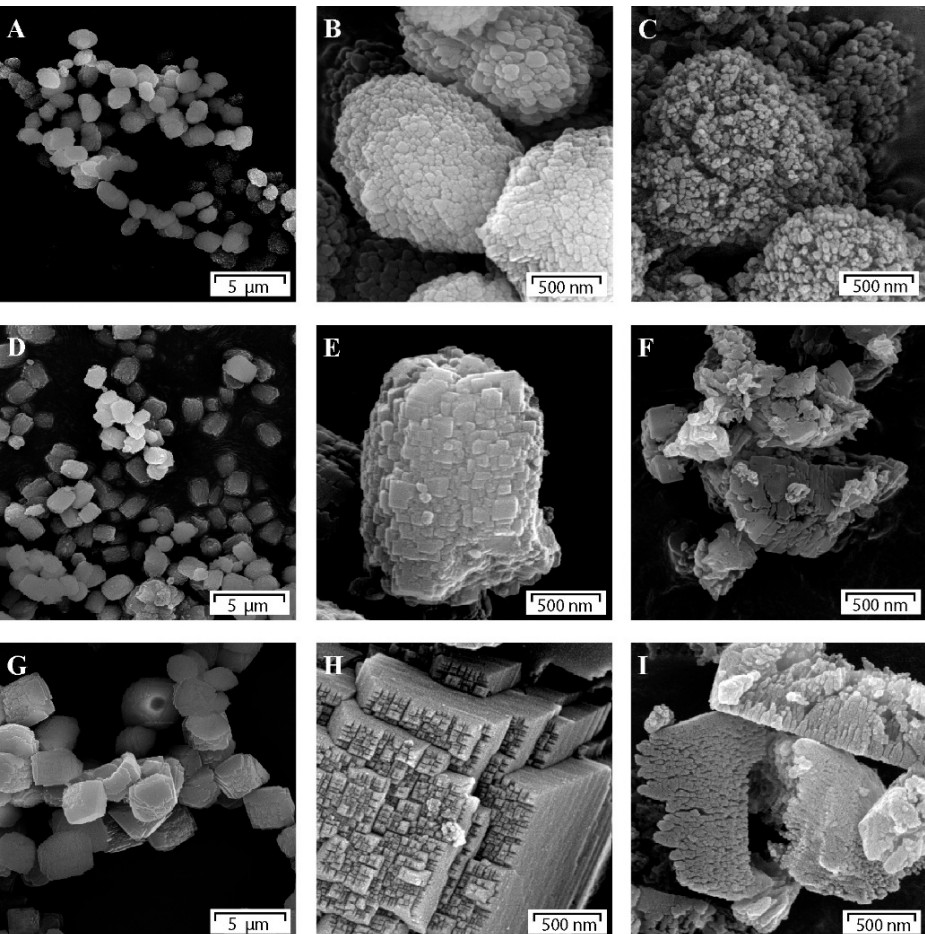

**Figure 9.** SEM images: (**A**) ZSM-5 standard; (**B**) ZSM-5 standard; (**C**) ZSM-5-at(60); (**D**) ZSM-12-(105) standard; (**E**) ZSM-12-(105) standard; (**F**) ZSM-12-(105)-at(60); (**G**) beta standard; (**H**) beta standard and (**I**) beta-at(120) [106]. Copyright (2011), with permission from Elsevier.

According to Svelle [107], the number of intergrowths in the ZSM-5 crystals determines the desilication characteristics. A large amount of intergrowth reduces the importance of the Si/Al ratio of the standard zeolite and leads to the formation of mesopores derived from the removal of defects. Figure 10 shows a schematic of the different mechanisms of mesopore formation, where small growths or defects require an ideal Si/Al ratio of 20–50 to introduce mesopores. Intermediate cases lead to a combination of the two mechanisms. In the case where many intergrowths are present, the Si/Al ratio is less important, and mesopores are formed mainly due to intergrowth/defect removal.

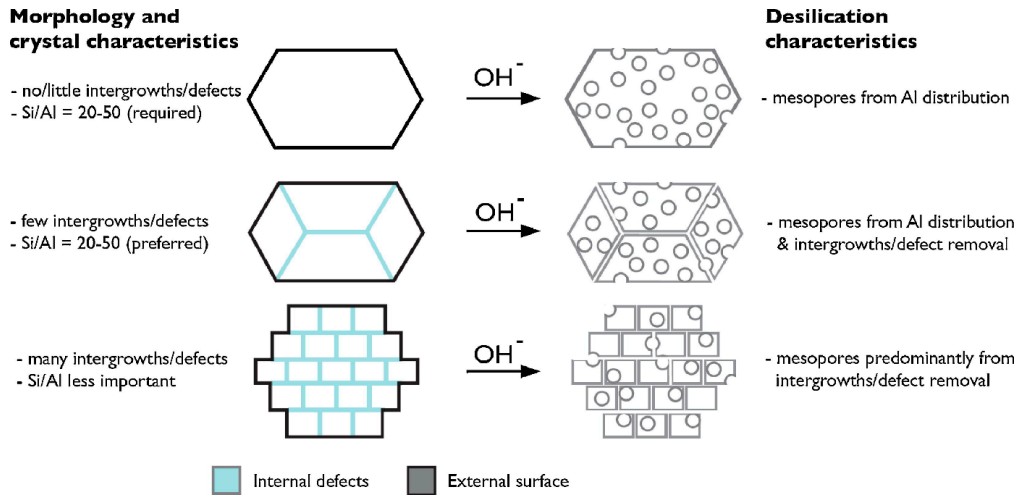

**Figure 10.** Schematic representation of different mechanisms of idealized mesopore formation [107]. Copyright (2011), with permission from Elsevier.

According to Verboekend [67], the optimal alkaline treatment strongly depends on the unique nature of each zeolite. For example, the properties that must be taken into account include the structure type, Si/Al ratio, Al distribution, crystal morphology and relative abundance of defects.

According to Tarach [89], compared to the previously reported desilication results for MFI, MOR and MTW zeolites, the relatively low stability of aluminum in the beta structure (caused by the presence of a high concentration of structural defects) negatively impacts the efficiency of the desilication process. High structural stability is a prerequisite for successful control of desilication, i.e., the coupling of mesoporosity development with the preservation of acidic properties.

Bi [108] successfully prepared the hierarchical zeolites H-ZSM-5 and Hβ (H-beta), which have interconnected micropores and mesopores, by alkaline treatment using different alkaline solutions ($Na_2CO_3$, NaOH, NaOH/TMAOH, NaOH/HCl). It was noted that the mesoporous surface areas of the hierarchical H-ZSM-5 zeolites were similar with different alkaline solutions. However, the surface area of the mesopores formed in the hierarchical Hβ zeolites was directly related to the basicity of the alkaline solution (more mesopores were produced in the NaOH solution). The main reason may be due to the different pore structures of the two zeolites. H-ZSM-5 contains 10-membered ring channels (0.51 nm × 0.55 nm and 0.53 nm × 0.56 nm), while Hβ contains 12-membered ring channels (0.66 nm × 0.67 nm and 0.56 nm × 0.56 nm). Hβ zeolite is more sensitive to alkaline post-treatment.

According to Al-Ani [105], a two-dimensional or three-dimensional pore system in some zeolites, for example, in medium-pore ZSM-5 (a zeolite with a three-dimensional pore system), may not be sufficient for the introduction of intracrystalline mesoporosity via the supramolecular modeling of ZSM-5 (alkaline treatment with NaOH/TPAOH + CTAB). These types of zeolites, with pores comprising 10-MR windows, are resistant to modification with surfactants; therefore, the potential routes for increasing their intracrystalline porosity are still under investigation. Additionally, according to the

authors, the mechanism of the mesostructuring process is not yet completely understood. A detailed understanding of the mesostructuring processes would serve as a general guide to allow targeted post-synthesis modification and, thus, the production of improved materials with a direct impact on new catalytic applications, particularly those involving transformations of bulky molecules.

According to Li [109], the introduction of mesopores by desilication into large crystals may be inefficient due to Al zoning. The desilication process is affected by the local aluminum content. Basic leaching of zoned crystals leads to the dissolution of the aluminum-poor parts of the crystals without creating mesopores in the aluminum-rich parts. The results of base leaching indicate that aluminum tends to propagate outward along with the crystallization (crystal growth) of ZSM-5.

### 3.1.5. Temperature and Treatment Time

Groen et al. [81] investigated the alkaline treatment (0.2 mol $L^{-1}$ NaOH) of ZSM-5 zeolite (Si/Al ratio = 37) with a focus on the evolution and optimization of the porous structure by varying the temperature (35–85 °C) and time (15–120 min) of treatment. They observed that the formation of mesoporosity during alkaline treatment is highly dependent on temperature and time. No substantial formation of mesopores was observed at temperatures up to 45 °C after 30 min of treatment. Only at 55 °C does some extra porosity begin to develop, and this porosity becomes substantial at 65 °C. Above 65 °C, the pore size distribution widens considerably, leading to a large volume of mesopores. A treatment temperature of 65 °C is considered optimal, combining extensive mesopore formation and almost completely preserved microporosity. Variations in treatment time in the range of 30–120 min at 65 °C lead to similar behavior but less spectacular changes. A longer treatment time yields a significantly wider pore size distribution, and a time of 30 min is the best to generate mesopores. The researchers concluded that the ideal temperature and time for the development of mesopores after the alkaline treatment of ZSM-5 are 65 °C and 30 min, respectively. Analysis of the influences of treatment temperature and time on the porous properties of the final material shows that the size and volume of mesopores can be adjusted to some extent by the optimization of these two variables.

Groen, Moulijn and Pérez-Ramirez [83] analyzed the effect of temperature and time in the alkaline treatment of ZSM-5 zeolite. The influence of temperature was studied in the temperature range 35–85 °C using 0.2 mol $L^{-1}$ NaOH solution as the alkaline medium and a treatment time of 30 min. At lower temperatures in the range of 35–45 °C, limited mesoporosity developed. The increase in the surface area of mesopores becomes significant after treatment at 55 °C. Accordingly, a strong increase in silicon leaching was observed. A maximum mesopore surface area of 240 m$^2$ g$^{-1}$ developed at a treatment temperature of 65 °C, which also led to a distinct decrease in microporosity. At higher temperatures, a decrease in the surface area of mesopores was observed. At these temperatures, excessive silicon extraction occurs, which leads to the formation of larger pores. Larger pores contribute strongly to the volume of mesopores but to a lesser extent to the surface area of mesopores. Consequently, a treatment temperature of 65 °C can be concluded to be optimal, considering the obtained surface area of mesopores in nanocrystalline zeolite crystals. The effect of time shows a similar trend. After exposure to NaOH at the ideal temperature of 65 °C for 15 min, the surface area of newly created mesopores already reached 180 m$^2$ g$^{-1}$, which was slightly lower than that after 30 min of treatment (240 m$^2$ g$^{-1}$). Similar to the effect of higher treatment temperatures, a longer treatment duration leads to a slight decrease in the surface area of mesopores, again due to excessive silicon dissolution and formation accompanied by larger pores.

Groen et al. [59] evaluated the effects of temperature (25–65 °C) and reaction time (10–60 min) in the alkaline treatment of beta zeolite with a Si/Al ratio = 35. They found that alkaline treatment at 25 °C hardly induced any new mesoporosity; the surface area of mesopores was only slightly larger than that of the standard sample; and treatment at 45 °C can be considered optimal in terms of combined micro- and mesoporous properties.

However, preferential silicon extraction can be substantially influenced by the treatment time, considering the large differences detected in the mesopore surface area of these samples and their progressively decreasing Si/Al ratios. At the ideal temperature (45 °C) for 10 min, mesopores are formed, and Brønsted acid sites remain. Extending the treatment time to 30 min causes the disappearance of /decrease in Brønsted acid sites and an increase in mesoporosity. Consequently, the marked influence of time on desilication was confirmed, not only in terms of porosity development but also in relation to the acidic properties associated with the presence of aluminum in the various materials. Shortening the treatment time may help retain the microporous and crystalline properties of purely microporous zeolite.

Santos et al. [110] studied the effect of time and temperature on the generation of secondary porosity in beta zeolite. The effect of time was evaluated using NaOH solution (0.2 mol L$^{-1}$) at 65 °C for a time period of 10–240 min. They observed that the formation of secondary porosity occurred under the selected experimental conditions, but the decrease in the microporous volume was drastic. The microporous volume decreased after a short treatment period of 20 min and then gradually increased. The mesoporous area initially increased with reaction time, reached a maximum value of 492 m$^2$ g$^{-1}$ in 20 min and then exhibited a relatively small reduction. The mesoporous area plateaued at 413 m$^2$ g$^{-1}$ after 240 min of treatment. The interesting results for beta zeolite as a function of treatment time led to the study of mesopore formation under a broader range of experimental conditions. Thus, the reaction temperature was varied from 30 to 100 °C, with the time set at 240 min. The mesopore area first increased with treatment temperature, reaching 499 m$^2$ g$^{-1}$ at 50 °C and then decreasing to 267 m$^2$ g$^{-1}$ at 100 °C. The microporous volume first decreased compared to the untreated sample and then increased. The trends observed with increasing treatment temperature were similar to those observed with respect to time.

Shah et al. [111] investigated the effects of alkaline treatment on the microporosity and acidity of the ZSM-5 zeolite structure by changing the temperature of the desilication method. The desilication reaction was performed at different temperatures (40, 60 and 80 °C). They observed that as the desilication temperature increased, the mesoporosity rate increased slightly after more Si was extracted. With increasing desilication temperature, the number of Lewis acid sites increased with the decreasing number of Brønsted acid sites. With increasing desilication temperature, a greater increase in pore size, acidity and mesoporosity was observed.

The temperature of the alkaline treatment is the key parameter in the generation of secondary porosity in zeolites with a well-defined structure. The pore size and microporosity distribution of zeolite can be controlled through temperature and time. According to the analyzed studies, it can be stated that temperature has a greater effect than time on the generation of secondary porosity.

## 4. Applications of Hierarchical Zeolites Prepared by Alkaline Treatment

The many efforts to obtain hierarchical zeolites with high thermal and hydrothermal stability and a high surface area without substantially losing their high micropore volume, acidity and crystallinity are mainly inspired by the industrial application of these materials in important reactions, for example, in catalysis, adsorption, optics, biomedicine and energy [108,109].

According to Hoff [34], the introduction of mesopores through alkaline treatment represents a possible strategy to improve intracrystalline diffusion and mass transport to promote the production of aromatics over unwanted coke formation.

According to Přech et al. [112], hierarchical zeolites exhibit improved the accessibility of active sites and faster mass transport and are generally more resistant to coke deactivation. Therefore, they often show higher catalytic activity than conventional zeolites, particularly in reactions that suffer from steric and/or diffusional limitations. In addition, the secondary porosity provides an ideal space for the incorporation and grafting of

other components and phases, opening a wide diversity of routes for the preparation of multifunctional materials.

García-Martínez et al. [15] studied fluid catalytic cracking (FCC) catalysts made from mesoporous zeolite Y, which showed significantly improved selectivity in product yields (more transportation fuels, i.e., gasoline and LCO, and less coke, dry gases and uncracked bottoms). Mesoporous zeolites are not only superior FCC catalysts but also ideal materials for a wide variety of other applications where slow diffusion is a limiting factor in the reaction process, e.g., hydrocracking, catalytic pyrolysis of biomass, catalytic enhancement of bio-oil, transesterification of vegetable oil, conversion of methanol into hydrocarbons, water treatment and less energy-intensive adsorptive separations.

According to Vu [113], the introduction of mesoporosity improves the performance of zeolite ZSM-5 in triglyceride cracking under FCC conditions. Compared to commercial ZSM-5, which is known as the best zeolite-type catalyst for triglyceride cracking, the hierarchical ZSM-5 catalyst increases conversion, gasoline and light olefin yields by approximately 14%, 10% and 30%, respectively. The superior catalytic performance of hierarchical ZSM-5 catalysts stems from the improved accessibility and mass transfer provided by the created mesoporosity and simultaneous maintenance of the intrinsic catalytic properties of ZSM-5. This allows hierarchical ZSM-5 zeolites to effectively convert triglycerides, regardless of the degree of unsaturation, into gasoline hydrocarbons and light olefins.

Ding et al. [114] investigated the effects of hierarchical H-ZSM-5 zeolites on the improvement of aromatic hydrocarbon yields during the fast catalytic pyrolysis (FCP) of waste cardboard (WCB). The carbon yield of BTX using the hierarchical zeolite increased by up to 82% compared to that of the standard H-ZSM-5 zeolite, resulting in an increase of up to 44% in aromatic carbon yield. There was a decrease in the carbon yield of sugars, carbonyl compounds and other oxygenated compounds. This improvement in product yields was attributed to the generated mesoporosity, which shortened the diffusion path length of molecules, and the increase in weak acid sites contributed to the improved selectivity of hierarchical H-ZSM-5 zeolites for BTX. Thus, the coke yield decreased due to the increase in pore size and improved diffusion performance.

McGlone et al. [90] investigated the [4 + 2] Diels–Alder cycloaddition of 2,5-dimethylfuran with ethylene using a series of hierarchical H-ZSM-5 catalysts synthesized by alkaline treatment. An increase in conversion was observed for all hierarchical materials compared to untreated zeolite, and increases in temperature and ethylene pressure significantly improved both the conversion of dimethylfuran and the selectivity to p–xylene due to easier desorption from the zeolite surface and the increased cycloaddition rate, respectively. Additionally, according to the authors, a compromise between acidity and mesoporosity was considered the key to increasing the activity and maximizing the selectivity in the production of p-xylene from 2,5-dimethylfuran.

According to Chen, Xiong and Tao [115], mesoporous ZSM-5 prepared by alkaline treatment proved to be an efficient catalyst for the hydrolysis of cellulose in ionic liquid (IL), providing a high yield of reducing sugars. It was demonstrated that mesoporous ZSM-5 had 76.2% cellulose conversion and a 49.6% total reducing sugar (TRS) yield. In comparison, conventional ZSM-5 showed only 41.3% cellulose conversion with 33.2% TRS yield. The results indicated that the important role of mesopores in zeolites in increasing the TRS yield might be due to the diffusional alleviation of cellulose macromolecules. It was found that IL could enter the internal channel of mesoporous ZSM-5 to promote the generation of $H^+$ from Brønsted acid sites, which facilitated hydrolysis. In addition, mesoporous ZSM-5 showed excellent reuse for catalytic cycles, which is promising for practical applications in cellulose hydrolysis.

Zhang [93] studied butene oligomerization using a series of new types of hierarchical H-ZSM-5 zeolite catalysts. It was demonstrated that alkaline treatment could effectively modify the acidity properties and hierarchical structure of the H-ZSM-5 catalyst. The results showed that hierarchical catalysts with interconnected open mesopores, smaller

crystal sizes and adequate acidity have a longer shelf life during butene oligomerization. A butene conversion rate of approximately 99% and C8+ selectivity of 85% were obtained at 12 h. Thus, an appropriate hierarchical catalyst can satisfy the requirements of the oligomerization process and has the potential to be used as a replacement for commercial ZSM-5 catalysts.

According to Tang [94], higher selectivity for aromatic hydrocarbons can be obtained in the FCP of lignin using the hierarchical zeolite catalyst H-ZSM-5. Alkaline treatment improved the catalytic performance of H-ZSM-5 zeolite for cracking bulky oxygenates released from lignin (such as guaiacol, syringol and their derivatives from lignin pyrolysis) to produce aromatic hydrocarbons.

According to Tanaka [95], the structure of hierarchical H-ZSM-5 zeolites modified with phosphorus by sequential alkaline/acid treatment reduced the residence time of light olefin products within the catalyst pores in methanol-to-olefin reactions, leading to a longer catalyst lifetime and higher methanol conversion and propylene selectivity.

Rac et al. [116] studied the possibility of improving the drug (atenolol, sodium diclofenac and salicylic acid) adsorption capacity of hierarchical ZSM-5 by using an alkaline treatment. Among the drugs tested, atenolol and diclofenac were most effectively adsorbed onto the hierarchical ZSM-5 zeolite. It was shown that, in the case of atenolol, the superior adsorption capacities of the hierarchical ZSM-5 may be even more pronounced at low initial concentrations. Therefore, the formation of mesopores can significantly improve the accessibility of active sites within the ZSM-5 structure.

Zhao [117] applied hierarchical H-MOR zeolite in the selective synthesis of ethylenediamine (EDA) via condensation amination of monoethanolamine (MEA) for the first time. It was observed that the diffusion conditions and the reactivities of the catalysts were improved by the generated secondary porosity and resulted in excellent catalytic performance under relatively mild reaction conditions. The MEA conversion was 52.8%, and the selectivity for EDA increased to 93.6% (according to the authors, this was the highest value ever reported). In addition, hierarchical H-MOR showed excellent catalytic stability in the selective synthesis of EDA. According to the authors, in the future, researchers can use hierarchical H-MOR as a support and still modify the structural and acidic properties of the catalyst, which will be one of the most promising methods to overcome the disadvantages of conventional methods of selective synthesis of EDA.

Our research group recently published an article on the application of hierarchical ZSM-5 zeolite in the catalytic cracking of polyethylene [118]. The hierarchical zeolite ZSM-5 was synthesized by alkaline treatment with NaOH in the presence and absence of CTAB using conventional processes of electric heating and microwave irradiation. It was observed that both forms of heating, regardless of the presence or absence of CTAB, efficiently formed secondary porosity in ZSM-5. The hierarchical samples exhibited efficient degradation of LDPE with a lower degradation temperature. This improvement in LDPE cracking is due to the introduction of secondary porosity, which consequently provides greater accessibility of LDPE molecules to the acid sites of hierarchical zeolites.

Smoliło-Utrata et al. [119] showed that modification of zeolite HY via alkaline treatment/desilication with vanadium impregnation could be an effective method to adjust the lattice oxygen basicity (a key parameter that plays a particularly important role in the process of oxidative dehydrogenation–ODH). The significantly increased mesopore surface ensures the binding of vanadium species to the silanol groups and the formation of the isolated $(SiO)_2(HO)V=O$ and $(SiO)_3 V=O$ sites or highly dispersed polymeric forms located in the zeolite. The higher basicity of lattice oxygen in deSi V-HY compared to V-HY, resulting from the presence of the Al-rich shell, aids in the activation of the C−H bond and greater selectivity to propylene.

Table 3 shows the summary of applications of hierarchical zeolites pre-pared by alkaline treatment cited in these items.

**Table 3.** Summary of applications of hierarchical zeolites prepared by alkaline treatment cited in these items.

| Hierarchical Zeolites | Applications | Reference |
|---|---|---|
| Y | Fluid catalytic cracking (FCC) | 2012 [15] |
| ZSM-5 | Triglyceride cracking | 2014 [113] |
| H-ZSM-5 | Fast catalytic pyrolysis (FCP) of waste cardboard | 2017 [114] |
| H-ZSM-5 | Diels–Alder cycloaddition of dimethylfuran and ethylene | 2018 [90] |
| ZSM-5 | Hydrolysis of cellulose in ionic liquid | 2018 [115] |
| H-ZSM-5 | Butene oligomerization | 2018 [93] |
| H-ZSM-5 | Catalytic pyrolysis of lignin | 2019 [94] |
| H-ZSM-5 | Methanol-to-olefin | 2020 [95] |
| ZSM-5 | Removal of pharmaceutically active substances | 2020 [116] |
| H-MOR | Selective synthesis of ethylenediamine via condensation amination of monoethanolamine | 2020 [117] |
| ZSM-5 | Polyethylene catalytic cracking | 2021 [118] |
| V-HY | Adjust the lattice oxygen basicity (oxidative dehydrogenation–ODH). | 2022 [119] |

## 5. Outlook and Final Considerations

Alkaline treatment has been proven to be an effective, inexpensive, versatile and simple post-synthesis procedure to generate secondary porosity in a wide variety of zeolite structures.

However, a disadvantage of alkaline treatment is that the use of a base alone can generate a wide range of mesopores and decrease crystallinity and acidity, which is undesirable.

The use of PDAs can help to overcome this limitation. The use of external PDAs allows the porosity, acidity and composition of hierarchical zeolites to be controlled. The presence of PDA on the outer surface induces partial protection that controls the dissolution process and leads to the formation of more uniform intracrystalline mesopores.

However, the use of PDAs during alkaline treatment can cause simultaneous desilication/reassembly. In this case, there is no loss of crystallinity, and a high surface area is obtained, making the application of this procedure even more desirable.

Thus, the careful choice of alkaline treatment conditions, such as the Si/Al molar ratio, alkaline agent, zeolite structure and treatment temperature and time, can lead to the realization of a hierarchical structure with preserved zeolite intrinsic properties.

Hierarchical zeolites synthesized by alkaline treatment have exceptional properties, such as improved accessibility of active sites, faster mass transport, a lower deactivation rate and better conversion capacity, thus making them promising for application in various industrial reactions.

However, to synthesize a specific hierarchical zeolite, it is necessary to carefully research the best alkaline treatment conditions to determine the ideal base to stabilize the zeolite structure during treatment. It is expected that this approach will be useful in achieving hierarchical forms of other zeolites by using an alkaline treatment.

**Author Contributions:** Conceptualization, S.B.C.P. and V.P.S.C.; methodology, D.S.O. and R.B.L.; formal analysis, V.P.S.C.; investigation, D.S.O. and R.B.L.; resources, S.B.C.P.; writing—original draft preparation, D.S.O. and R.B.L.; writing—review and editing, S.B.C.P. and V.P.S.C.; visualization, V.P.S.C.; supervision, S.B.C.P.; project administration, V.P.S.C.; funding acquisition, S.B.C.P. All authors have read and agreed to the published version of the manuscript.

**Funding:** This research was funded by CAPES, grant number 88887.370976/2019-00.

**Data Availability Statement:** Not applicable.

**Conflicts of Interest:** The authors declare no conflict of interest.

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
