# Peer review of "Hierarchical Zeolite Synthesis by Alkaline Treatment: Advantages and Applications"

_catalysts, doi:10.3390/catal13020316_

Round 1

Reviewer 1 Report

The review presented enlarges study based on transformed uniform zeolitic micropores into different hierarchical structures in different alkaline media. The review contents valuable scientific information of experimental preparation, Characterization and some of applications. I recommend publishing this review in catalysts after minor revision.  

The keywords are very generic

In the table 1 i recommend you to insert references for the presented methodologies

In the figure 1 you presented a microscope images, details and references are needed!

You mentioned table 2 only one time, which makes a gap between the table and the text of discussion for the reader.  Try to mention the table in the text in the suitable form to be more useful. Also, i recommend you to put the year for each!

In the part of "Applications of hierarchical zeolites prepared by alkaline treatment " i recommend you to introduce a table to be more useful for the reader  

Author Response

Thanks for the suggestions. We replay your suggestions below:

1) The keywords are very generic.

Yes, we put new keywords: 

Keywords: Hierarchical zeolites; Alkaline treatment; Porous materials; Synthesis strategies; Post-synthesis methods.

2) In the table 1 i recommend you to insert references for the presented methodologies.

Yes, we do it.

3) In the figure 1 you presented a microscope images, details and references are needed!

Yes, this Figure 1 are made by us. We put details: 

Figure 1. Method of desilication by alkaline treatment (SEM image of the ZSM-5 sample performed in TESCAN equipment, model MIRA3 FEG-SEM, magnification 2 µm).

4) You mentioned table 2 only one time, which makes a gap between the table and the text of discussion for the reader.  Try to mention the table in the text in the suitable form to be more useful. Also, i recommend you to put the year for each!

Yes, we put the year of each reference and put a phrase to connect the table to the text.

5) In the part of "Applications of hierarchical zeolites prepared by alkaline treatment " i recommend you to introduce a table to be more useful for the reader .

Yes, we put a table that summarizes the applications.

Reviewer 2 Report

In my opinion, the manuscript is a valuable review collecting many information on the production, properties and application of hierarchical zeolites. Reading it may be helpful to everybody who works in this subject. The authors presented the literature data in very well organized manner.

I propose to accept it with small modifications.

Even though the authors took into account a lot of publication, I propose to add information in several other papers. Most of them concern the desilication of zeolite Y what produces zeolites of good porosity, good acidity and good catalytic properties. Moreover zeolite Y which was ultrastabilized and next desilicated contains hydroxyls the most acidic in all the chemistry of zeolites. The list of the papers concerning desilicated zeolite Y is given below.

1.       K.P. de Jong, et al. Angew. Chem. Int. Ed. 49 (2010) 10074

 2.   J. Van Aelst et al  J. Phys. Chem. C., 118 (2014) 22573

3  D. Verboekend et al.  Chem. Soc. Rev. 45 (2016) 3331-3352.

4.   D. Verboekend et al; Adv. Funct. Mater. 23 (2013) 1923

5.  V. Rac et al. Microp. Mesop. Mater.194 (2014) 126

6. M. Gackowski, et al., Molecules, 2020, 25, 31

7. M. Gackowski, et al. Molecules, 2020, 25, 1044

Very valuable results were obtained with desilicated zeolite of mazite type (M. Gackowski, et al. Appl. Catal. A, 2019, 578, 53). I propose to take them into account too.

Lines 436-438. The authors say that „ alkalinity of TPAOH is very weak”. This is not true. TPAOH is relatively strong base, however it acts as a template, helping to repair the crystal structure of zeolite, as stated by the authors in the  next sentence.

Author Response

Thanks for the suggestions, we answer each one bellow:

1) Even though the authors took into account a lot of publication, I propose to add information in several other papers. 

  1. K.P. de Jong, et al. Angew. Chem. Int. Ed. 49 (2010) 10074
  2.  J. Van Aelst et al  J. Phys. Chem. C., 118 (2014) 22573
  3.  D. Verboekend et al.  Chem. Soc. Rev. 45 (2016) 3331-3352.
  4. D. Verboekend et al; Adv. Funct. Mater. 23 (2013) 1923
  5. V. Rac et al. Microp. Mesop. Mater.194 (2014) 126
  6. M. Gackowski, et al., Molecules, 2020,25, 31
  7. M. Gackowski, et al. Molecules, 2020,25, 1044.

Yes, the papers 1 and 4 are yet in the text. The others we put now.

2) Very valuable results were obtained with desilicated zeolite of mazite type (M. Gackowski, et al. Appl. Catal. A, 2019, 578, 53). I propose to take them into account too.

Yes, we put it in the text.

3) Lines 436-438. The authors say that „ alkalinity of TPAOH is very weak”. This is not true. TPAOH is relatively strong base, however it acts as a template, helping to repair the crystal structure of zeolite, as stated by the authors in the  next sentence.

Yes, we change the phrase to: 

Treatment with TPAOH did not have a significant effect on the introduction of mesoporosity into H-ZSM-5 because TPAOH acted as a template, helping to repair the crystal structure of zeolite

Reviewer 3 Report

Oliveira et al. review Hierarchical Zeolite Synthesis as announced in the title of this manuscript.

The review helps scientist interested in this topic to obtain a relevant and timely overview.

My impression is that this manuscript is useful and can be published as it is.

Author Response

Thanks for your words. We are happy that you like the paper.

Reviewer 4 Report

This paper presents various examples of applying new reactions by modifying microporous zeolites into micro/mesoporous zeolites with simple alkaline treatment. It can be considered as an excellent review paper outlining various ways to extend zeolite applications to a wider range in the future. It can be considered publishable in the present state.

Author Response

(The authors gave the same response as above.)
